# The Appearance of Antiphage Antibodies in Sera of Patients Treated with Phages

**DOI:** 10.3390/antibiotics14010087

**Published:** 2025-01-14

**Authors:** Marzanna Łusiak-Szelachowska, Beata Weber-Dąbrowska, Maciej Żaczek, Ryszard Międzybrodzki, Andrzej Górski

**Affiliations:** 1Bacteriophage Laboratory, Hirszfeld Institute of Immunology and Experimental Therapy, Polish Academy of Sciences (HIIET PAS), 53-114 Wrocław, Poland; beata.weber-dabrowska@hirszfeld.pl (B.W.-D.); maciej.zaczek@hirszfeld.pl (M.Ż.); ryszard.miedzybrodzki@hirszfeld.pl (R.M.); andrzej.gorski@hirszfeld.pl (A.G.); 2Phage Therapy Unit, Hirszfeld Institute of Immunology and Experimental Therapy, Polish Academy of Sciences (HIIET PAS), 53-114 Wrocław, Poland; 3Department of Clinical Immunology, Transplantation Institute, The Medical University of Warsaw, 02-006 Warsaw, Poland; 4Department of Immunology, The Medical University of Warsaw, 02-006 Warsaw, Poland

**Keywords:** antiphage antibody, duration of phage therapy, phage neutralization of sera, phage therapy

## Abstract

**Background**: Bacteriophages are neutralized by the sera of patients undergoing phage therapy (PT), particularly during local or concomitant local and oral phage administration in bone infections, soft tissue infections, or upper respiratory tract infections. **Methods:** The antiphage activity of the sera (AAS) level of 27 patients with bacterial infections such as bone infections, soft tissue infections, or upper respiratory tract infections undergoing PT was performed using the plate phage neutralization test. **Results:** Our preliminary results suggest that high levels of antiphage antibodies appear late in the treatment period, at the earliest in the 3rd–8th week of PT. Patients with bone infections treated locally with the *S. aureus* phage Staph_1N and patients with upper respiratory tract infections administered locally and orally with the *S. aureus* phage Staph_A5L had elevated levels of antiphage antibodies in sera during PT. In parallel to these results, it has been shown that a strong antiphage humoral response does not prevent a positive outcome of phage treatment. **Conclusions:** The earliest time point at which high levels of antiphage antibodies in sera appear during local and oral PT is day 21 of therapy. We noticed that the high level of antiphage antibodies in sera occurring during local or both local and oral PT was correlated with the type of infection and phage type.

## 1. Introduction

Researchers are becoming increasingly interested in finding new alternative methods of combating antibiotic-resistant bacteria, including the use of phage therapy (PT) [1,2,3]. Although the antibacterial effect is the most recognized feature among scientists, the diversity of bacteriophage properties goes beyond this pattern. They may interact with the immune system, which in turn may influence the induction of antiphage antibodies in both animals and humans. The clinical significance of this phenomenon remains unclear [4,5,6,7]. Antiphage antibodies can neutralize phage activity, thereby reducing the ability of phages to destroy bacteria.

Our previous studies have shown that the level of antiphage activity of sera (AAS) calculated as the rate of phage inactivation (K) in patients undergoing PT depends on several factors such as the route of phage administration, phage type, and the dose of the phage preparation (Figure 1) [6]. Oral and rectal administration of phages are considered the least immunogenic, whereas the combination of both local and oral administration is more immunogenic. Purified phage preparations with higher phage titer may be more immunogenic in comparison to phage lysates [6]. The correlation between the higher phage titer and the higher antibody level is not a rule, and the degree of phage purification may also play an essential part. Żaczek et al. [7] showed that sometimes lysates are more immunogenic (despite lower titers) than purified preparations, which was explained by the influence of impurities on the higher antibody levels. Antibody responses to phage administration may also depend on the patient’s immune status, which is clinically relevant as approximately 50% of them are immunodeficient. Moreover, a positive correlation was noted between the level of IgG and IgM antiphage antibodies and the K rate in sera of patients treated with phages [7]. Interestingly, we demonstrated that the level of AAS was not correlated with the outcome of PT [6].

Nick et al. [8] and Le et al. [9] confirmed our observations. The authors revealed that clinical improvement in PT was observed in the case of lung or urinary tract infection despite the antiphage antibody response. Other researchers also confirmed that PT may be successful despite the induction of antiphage antibodies in lung and cystic fibrosis diseases [10,11,12].

The current study aims to determine when the earliest time point is for the high levels of AAS to appear during both local and oral PT of bone infections, soft tissue infections, or upper respiratory tract infections.

## 2. Results

As mentioned above, a total of 27 patients with bone infections, soft tissue infections, or upper respiratory tract infections were treated with *S. aureus* phages Staph_1N or Staph_A5L either locally or as a concomitant local and oral treatment (Table 1 and Table 2 respectively). Patients underwent phage therapy (PT) until week 8 after the administration of the first dose. Prior to the PT, patients had low rates of phage inactivation (K = 0.00–0.52) in sera. After 2–2.5 weeks of PT administered locally, patients had low K = 0.29–0.67, whereas in weeks 3–8, their K was classified as low or high with values up to K = 160.08 in sera (Table 1). After 2–2.5 weeks of PT applied locally and orally, patients had low K = 0.10–2.59, whereas in weeks 3–8, their sera samples yielded low or high K up to K = 75.12 (Table 2). The earliest time point when high levels of antiphage antibodies in sera were observed fell on weeks 3–8 of PT.

High levels of AAS during local PT were detected in three patients with bone infections and in one patient with a soft tissue infection (Figure 2). Three patients in this group used *S. aureus* phage Staph_1N, and one patient used *S. aureus* phage Staph_A5L. In the second group of patients who applied phages locally and orally, high levels of AAS were detected in one patient with bone infection and in two patients with upper respiratory tract infection (Figure 3). All patients in this group were administered *with S. aureus* phage Staph_A5L.

Twice as few patients had K > 18 during or after local and oral treatment (25%) than those treated only locally (45.4%), which may be the result of a different route of phage administration to patients, as well as the type of infection and phage type. As for the treatment outcome, 27.3% of patients treated locally with phages had positive clinical responses to PT (A–C). A higher percentage of positive outcomes of PT 43.7% (A–C) was observed in patients treated with phages both locally and orally.

## 3. Discussion

The above-mentioned studies confirm our previous findings [6], which are of high importance for phage therapy: the idea that bacteriophages are neutralized during PT and that the level of antiphage antibodies does not affect the result of PT, which is still poorly understood. Recent studies from 2022 by Nick et al. [8] and from 2023 by Le et al. [9] confirm the lack of influence of a strong antiphage antibody response in sera during PT, in lung or urinary tract infection, on the outcome of PT.

In this article, the time of occurrence of high K level in sera in local and a combination of both local and oral PT was compared. The same staphylococcal phages (two types of phages Staph_1N or Staph_A5L) were used in bone infections, soft tissue infections, or upper respiratory tract infections. Similarly to local PT, in combined local and oral PT high K levels in sera first occurred in weeks 3–8 of PT. The higher number of patients with K > 18 during or after local PT vs. local and oral PT may also result from the type of infection (bone infections) and use of the *S. aureus* phage Staph_1N. Our studies also indicated a positive clinical response in some patients with high AAS during or after local PT in soft tissue infection and bone infection who used *S. aureus* phage Staph_1N. Also, a clinical improvement was observed during local and oral PT in a patient with high AAS with bone infection who used *S. aureus* phage Staph_A5L.

Recent data derived from a murine model confirm our observations that two rounds of phage treatment are necessary to induce neutralizing antibodies [13].

In the studies of Le et al. [9], in the case of urinary infections partial sera inactivation of two *Klebsiella* phages out of 3 from the phage cocktail was observed in the 1st week during intravenous phage administration. Phage therapy was conducted for 4 weeks without concomitant use of antibiotics. A higher level of phage inactivation in the serum was observed on day 8 versus day 15. The reason for this phenomenon is unclear. Other studies by Nick et al. [8] focusing on lung infections demonstrated that the level of antiphage antibody in the serum to one mycobacteriophage out of 2 from a phage cocktail increased with time. A gradual increase in neutralization was observed from 3 to 500 days post-phage treatment administered intravenously with substantial inactivation after 242 days. The study indicated that the appearance of high levels of antiphage antibodies in the late phage treatment does not exclude the positive results of PT. Studies by Dan et al. also reported that the robust phage-specific immune response did not prevent clinical treatment success, as evidenced by the absence of pneumonia and microbiological tests [10]. A multidrug-resistant *Pseudomonas aeruginosa* in a lung transplant recipient was treated with adjunctive intravenous and inhaled PT. A patient was administered two phage cocktails AB-PA01 and its modified version AB-PA01-m1, as well as the United States Navy antipseudomonal phage cocktail successively until day 92. Neutralizing antibodies for two phage cocktails peaked at days 63 and 70.

However, a limited phage therapeutic efficacy was observed in a case of a pulmonary *Mycobacterium abscessus* infection with the antibody response to phage [14]. A patient with refractory lung disease was treated intravenously for six months with a three-phage cocktail. After two months high levels of IgM and IgG-neutralizing antibodies in phages were associated with clinical deterioration. An increase in the number of bacteria in sputum was detected.

Dedrick et al. presented the results of aerosolized phage therapy in a patient with refractory *M. abscessus* lung disease. Intravenous cocktails composed of three mycobacteriophages were administered over a six-month period, alongside a dedicated antibiotic regimen [15]. A transient decline in the *M. abscessus* burden, followed by a rebound in mycobacterial colony-forming units, was observed. The authors reported no changes in the phage and antibiotic susceptibility of *M. abscessus* isolates after eight months of nebulization. However, there was no substantial clinical effectiveness noted after six months of twice-daily intravenous and nebulized phage treatment. The study provides valuable insights into the interaction between phages and the immune system, particularly the role of IgA in sputum. While initial phage administration may face challenges due to weak neutralization by IgA, the potential for an enhanced immune response over time suggests that phage therapy could still be a viable treatment option, especially with individualized approaches that consider the unique immune responses of patients. Further research and clinical trials are warranted to optimize phage therapy protocols for mycobacterial infections.

Our previous studies have shown an increase in the level of IgM and IgG antiphage antibodies in sera of some patients with bone infections treated locally with staphylococcal MS-1 phage cocktail, while IgA antiphage antibodies in sera were low [7]. The authors observed two exceptions in this group of patients where IgA levels in sera were elevated. It is possible that higher levels of IgA antiphage antibodies could be detected in saliva instead of serum.

## 4. Materials and Methods

### 4.1. Patients

A total of 27 patients with bacterial infections such as bone infections, soft tissue infections, or upper respiratory tract infections underwent PT between 2017 and 2023 at the Phage Therapy Unit (PTU) of Hirszfeld Institute of Immunology and Experimental Therapy, Polish Academy of Sciences (HIIET PAS) in Wrocław, Poland. The patients were administered monovalent phage lysate *Staphylococcus aureus* Staph_1N or Staph_A5L (Table 1 and Table 2). Bacteriophage preparations were sourced from the Laboratory of Bacteriophages (HIIET) Wrocław. Phage lysates were prepared according to the procedure described in Żaczek et al. [7]. The titer of phage preparations varied between 10^6^ and 10^8^ plaque-forming units/mL (pfu/mL). Lytic phages were selected for treatment based on the phage typing procedure. Phages were used locally (Table 1) or as a combined local and oral treatment (Table 2) according to the protocol “Experimental phage therapy of drug-resistant bacterial infections, including MRSA infections” [16]. The local phage administration was performed two to three times daily (compress, fistula irrigation, spraying onto the skin, spraying wound, nasal spray, and sinus irrigation). The oral phage administration was performed using a 10 mL phage preparation two to three times daily, at least 30 min before meals. About 10 mL dihydroxy aluminium sodium carbonate solution (68 mg/mL) was applied up to 20 min before phage application to prevent phage inactivation by gastric juice.

The blood was collected before, during, and after PT. As previously described, patients underwent phage therapy (PT) according to the established experimental therapy protocol, and the number of visits ranged from 1 to 11 depending on the course of therapy [7]. Despite our efforts, it was impossible to collect samples according to clinical trial protocols due to limitations associated with patient availability. The blood was centrifuged at 1500× *g* for 10 min and separated sera samples were collected into cryovials and stored at −70 °C. AAS was performed immediately after obtaining sera samples. The AAS studies were performed in line with the approval of the Bioethics Committee of the Wrocław Medical University (Poland). All patients gave written informed consent to participate in these studies.

### 4.2. Bacteriophages

*S. aureus* bacteriophages vB_SauH_A5L18349 and vB_SauH_1N/80 from the HIIET PAS, Wrocław, Poland phage collection were used in the study. Both phages represent polyvalent, obligatory lytic phages from serological group D (with 93–99.8% genome homology inside the group). According to ICTV taxonomy, they belong to the family *Herelleviridae*, subfamily *Twortvirinae*, and genus *Kyavirus G1.* Their genomes consist of double-stranded DNA, and they do not exhibit the ability to package foreign DNA into the virion. Additionally, they do not encode any homologs of known virulence factors or determinants of antibiotic resistance [17]. These phages demonstrate a broad host range and lytic spectrum, effective against 73–84% of tested strains [17,18,19] and our unpublished data]. The phages used in the study are not highly specific, they both demonstrated lytic activity against staphylococcal strains belonging to species: *S. aureus* MSSA, MRSA, MLSB, *Staphylococcus K(-)* MRSNS, MSCNS, *Staphylococcus haemolyticus*, *Staphylococcus epidermidis,* and *Staphylococcus lugdunensis*. Additionally, phage Staph_A5L—to *Staphylococcus hominis* and *Staphylococcus warneri*, but phage Staph_1N—to *Staphylococcus epidermidis* MRSE and *Staphylococcus sciuri* strains.

### 4.3. Plate Phage Neutralization Test

The AAS level of patients undergoing PT was calculated using the plate phage neutralization test as described earlier [6,20]. Briefly, the phage was used at a concentration of 1 × 10^6^ pfu/mL and the sera samples were diluted from 1:10 up to 1:1500. A total of 50 μL of phage was mixed with undiluted serum and each of the prepared serum dilutions (450 μL). The mixture was incubated at 37 °C for 30 min. After incubation, the mixture was diluted 100-fold with cold-enriched broth (5.4 g of enzymatic hydrolysate of casein, 4 g of Bacto Peptone, 3.5 g of NaCl, 1.7 g of yeast extract, 0.4 g of beef extract and 1000 mL of distilled water). The phage titer was determined using the double-agar layer method described by Adams [21]. A total of 100 μL of the mixture and 200 μL of bacterial culture were added to 3 mL of semi-liquid agar (0.7%) at 45 °C and the mixture flowed out on the agar plate (5 g of NaCl, 3 g of Na_2_HPO_4_, 10 g of enzymatic hydrolysate of casein, 3 g of beef extract per litre, and 1.2% of agar). After 8 h of incubation at 37 °C, plaques were detected. AAS was assessed as the rate of phage inactivation K.$$K=2.3\cdot \left(\frac{D}{\bar{T}}\right)\cdot \log\left(\frac{P0}{Pt}\right)$$
where D is the reciprocal of the serum dilution, T is the phage-serum reaction time (30 min.), P0 is the phage titer at the beginning of the phage-serum reaction and Pt is the phage titer after time T of the phage-serum reaction.

A K rate of less than 5 was determined as a low level of phage inactivation, a K of between 5 and 18 as a medium level of phage inactivation, and above 18 as a high level of phage inactivation.

### 4.4. Categories of the Results of PT

The outcome of PT was described according to Międzybrodzki et al. [16].

Categories A–C were assigned as positive responses to PT:A—Pathogen eradication and/or recovery (eradication confirmed by the results of bacterial cultures. Recovery refers to wound healing or complete subsidence of the infection symptoms);B—Good clinical result (almost complete subsidence of some infection symptoms, together with significant improvement in the patient’s general condition after completion of PT);C—Clinical improvement (discernible reduction in the intensity of some infection symptoms after completion of PT to a degree not observed before PT, when no treatment was used).

Categories D–G were assigned as inadequate responses to PT:D—Questionable clinical improvement (reduction in the intensity of some infection symptoms to the degree that could also be observed before PT);E—Transient clinical improvement (reduction in the intensity of some infection symptoms observed only during the application of phage preparations and not after the termination of PT);F—No response to treatment (lack of reduction in the intensity of some infection symptoms observed before PT);G—Clinical deterioration (of symptoms of infection at the end of PT).

## 5. Conclusions

In some patients, high levels of antiphage antibodies in sera may appear as early as in the 3rd–8th week of PT. This is especially visible during local or concomitant local and oral therapy in bone infections, soft tissue infections, or upper respiratory tract infections. The earliest time point, at which high levels of antiphage antibodies in sera appear during local and oral PT is day 21 of therapy. This phenomenon does not weaken the outcome of PT. Additionally, we observed that the high level of AAS during local PT was correlated with the type of infection and phage type, especially with bone infection and treatment with the *S. aureus* phage Staph_1N. Whereas, the high level of AAS during local and oral PT was observed especially in upper respiratory tract infection with the use of the *S. aureus* phage Staph_A5L. The studies should be continued to monitor the occurrence of neutralizing antiphage antibodies in patients undergoing PT. It is also important to develop studies to determine the level of IgM, IgG, and IgA antiphage antibodies in the sera of patients using PT as each class of antibodies appears at a different point in therapy and may shed new light on the patient’s immune status. Further research is necessary to explain in more detail the factors behind antibody production during PT. Given the contradictory results discussed in this paper and the importance of immunogenicity testing in all newly designed clinical trial protocols, our research is an important contribution to the further development of phage therapy in humans.

## Figures and Tables

**Figure 1 antibiotics-14-00087-f001:**
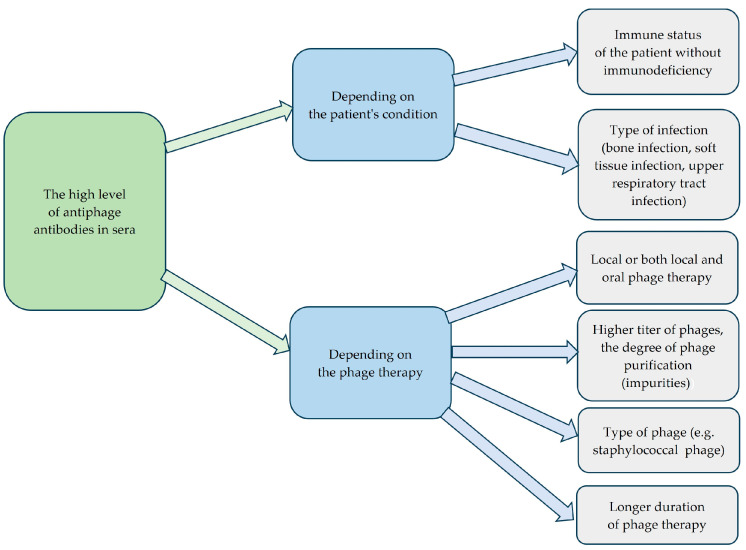
Factors influencing high levels of antiphage antibodies in sera.

**Figure 2 antibiotics-14-00087-f002:**
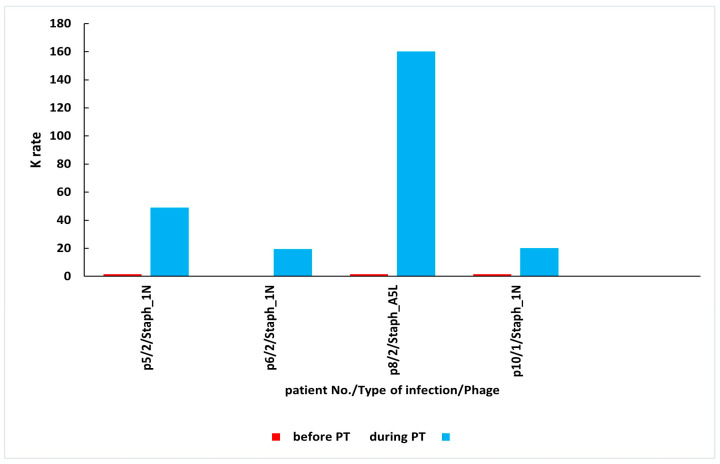
High levels of AAS during local PT. AAS—antiphage activity of sera; PT—phage therapy; K—rate of phage inactivation; 1—soft tissue infection; 2—bone infection. The absence of antiphage antibodies before PT in p6/2/Staph_1N. Patient numbers in the graph correspond to numbers in Table 1. Mean K ± SD before PT is 0.16 ± 0.17; mean K ± SD during PT is 62.15 ± 66.70.

**Figure 3 antibiotics-14-00087-f003:**
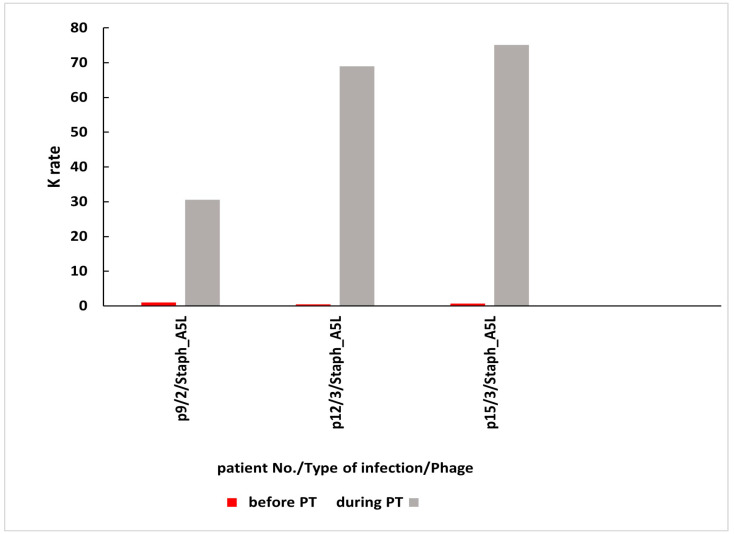
High levels of AAS during local and oral PT. AAS—antiphage activity of sera; PT—phage therapy; K—rate of phage inactivation; 2—bone infection; 3—upper respiratory tract infection. Patient numbers in the graph correspond to numbers in Table 2. Mean K ± SD before PT is 0.22 ± 0.26; mean K ± SD during PT is 58.21 ± 24.14.

**Table 1 antibiotics-14-00087-t001:** Patients treated locally with *S. aureus* phages.

Patient No.	Type of Infection	Phages Used in PT	K Before PT	K During PT	K After PT	Days of PT	Clinical Outcome of PT ^a^
1	soft tissue infection	Staph_1N	0.27	0.36	n.s.	14	D
2	bone infection	Staph_1N	0.15	0.29	n.s.	18	F
3	upper respiratory tract infection	Staph_1N	0.19	0.67	n.s.	18	F
4	bone infection	Staph_A5L	0.04	0.38	n.s.	21	F
5	bone infection	Staph_1N	0.05	48.86	n.s.	21	E
6	bone infection	Staph_1N	0.00	19.51	n.s.	28	D
7	bone infection	Staph_1N	0.02	0.02	0.02	28	C
8	bone infection	Staph_A5L	0.25	160.08	303.99	35	D
9	soft tissue infection	Staph_A5L	0.05	0.25	n.s.	56	D
10	soft tissue infection	Staph_1N	0.36	20.15	n.s.	56	C
11	bone infection	Staph_1N	0.02	143.12 (2.5 months after PT)	22.18 (1 year and 8 months after PT)	24	C
		Mean K ± SD	0.12 ± 0.11	35.79 ± 56.56			

Legend: K—rate of phage inactivation; PT—phage therapy; ^a^ Results A–C positive responses to PT; Results D–G inadequate responses to PT. n.s. no serum, SD—standard deviation. The K rate rose significantly (*p* = 0.005, Wilcoxon test) in patients receiving phages locally.

**Table 2 antibiotics-14-00087-t002:** Patients treated locally and orally with *S. aureus* phages.

Patient No.	Type of Infection	Phages Used in PT	K Before PT	K During PT	K After PT	Days of PT	Clinical Outcome of PT ^a^
1	soft tissue infection	Staph_1N	0.06	0.32	n.s	12	E
2	soft tissue infection	Staph_A5L	0.14	0.15	0.05	14	E
3	soft tissue infection	Staph_A5L	0.47	2.59	n.s.	14	B
4	soft tissue infection	Staph_1N	0.08	0.10	n.s.	14	B
5	upper respiratory tract infection	Staph_A5L	0.03	0.72	n.s.	14	C
6	soft tissue infection	Staph_1N	0.18	0.12	n.s.	17	B
7	upper respiratory tract infection	Staph_1N	0.02	1.42	n.s.	17	E
8	soft tissue infection	Staph_1N	0.02	0.27	0.01	18	A
9	bone infection	Staph_A5L	0.52	30.56	n.s.	21	C
10	bone infection	Staph_A5L	0.03	0.34	n.s.	24	D
11	soft tissue infection	Staph_A5L	0.26	2.62	n.s.	28	C
12	upper respiratory tract infection	Staph_A5L	0.02	68.94	28.04	30	E
13	upper respiratory tract infection	Staph_1N	0.34	0.53	n.s.	47	F
14	soft tissue infection	Staph_1N	0.46	0.58	n.s.	55	F
15	upper respiratory tract infection	Staph_A5L	0.11	75.12	n.s.	56	E
16	bone infection	Staph_1N	0.14	2.86	71.79 (1 month after PT)	35	D
		Mean K ± SD	0.18 ± 0.17	11.70 ± 23.93			

Legend: K—rate of phage inactivation; PT—phage therapy; ^a^ Results A–C positive responses to PT; Results D–G inadequate responses to PT. n.s. no serum, SD—standard deviation. The K rate rose significantly (*p* = 0.0007, Wilcoxon test) in patients receiving phages locally and orally.

## Data Availability

The data presented in this study are derived from personal patients’ medical records maintained at the Phage Therapy Unit of the Medical Centre as well as Bacteriophage Laboratory of the Institute of Immunology and Experimental Therapy Polish Academy of Sciences in Wrocław, Poland. Those data are not publicly available due to privacy and legal issues (The General Data Protection Regulation (EU) 2016/679 and Act on the rights of the patient and the Patient’s Rights Ombudsman from 6 November 2008).

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
