# Peer review of "The Appearance of Antiphage Antibodies in Sera of Patients Treated with Phages"

_antibiotics, 2025, doi:10.3390/antibiotics14010087_

Round 1

Reviewer 1 Report

Comments and Suggestions for Authors

I revised the manuscript. I added my comments to the manuscript.

Comments on the Quality of English Language

I revised the manuscript. I added my comments to the manuscript.

Author Response

Editor of Antibiotics                                                                    Wrocław, 20.12.24

Cover Letter

Dear Editor, we entirely agree with you that   "Self- citations may be valid if, for example, they are needed to understand the background and history of the work in question". This statement applies to our situation. We have been a pioneering phage therapy center in Europe with the largest number of patients treated and monitored for their antibody responses against phages. In fact, such responses are only rarely mentioned in publications related to phage therapy which often include only single cases. Moreover, our studies on factors influencing the intensity of antibody responses against phages during clinical phage therapy are quite unique. Therefore, when discussing the data presented in the current paper we had to refer to our earlier reports as no similar analyses had been published so far. Recent data derived from a murine model confirm our observations that two rounds of phage treatment are necessary to induce neutralizing antibodies (Berkson JD et al, Nature Com. 2024, 15:2993). This citation has now been added to the reference list. Our corrections are marked in red. Linguistic corrections from the reviewer 1 are marked in green.

English has been checked again.

Regards

  1. Łusiak-Szelachowska

Reviewer 1

Thank you very much for your opinion and suggestions.

The results and the English we improved according to reviewers suggestions.

We included comment in the legend of Figure 2 about the absence of antiphage antibodies in case 2/Staph_1N  before PT.

We added K in tables after PT in some patients. The studies were conducted in accordance with the experimental therapy. We could not obtain the results after therapy from all patients.

Instead of numbers, we have included types of infections in tables 1 and 2.

Linguistic corrections from the reviewer are marked in green.

Reviewer 2 Report

Comments and Suggestions for Authors

The effects of antibody produced by patients undergoing phage therapy (PT) to the outcome of treatment are still poorly known thus the results of this study could provide additional data on the topic. Especially, when PT is likely to be a major alternative for disease treatment to prevent resistance to antibiotics in the future. Results of the experiments performed by Authors showed that antiphage antibodies accumulate only late in the treatment period of patients with bone infections, soft tissue infections or upper respiratory tract infections. This could be the reason why the antibody seemed to not influence the positive outcome of PT although it did actually neutralize bacteriophages used in the treatment.  I only have several suggestions below.

Please add more details of the Results to the Abstract to improve its clarity and attract more Readers.

Lines   65, 67 & 97     Please write city and country names where these institutions located as some Readers may not familiar with them. Please only use abbreviations (e.g. HIIET) after their first mention.

Table 1                        Why only rates of phage inactivation (K) before and during PT were presented here while patient bloods were also collected (and tested?) after PT?

Lines 91-93                 Please also provide these legends to Table 1 as it is a separate table to Table 2.

Lines 105                    This statement needs reference(s).

Lines 116-117             Please explain how to prepare the media agar, or at least its composition.

Lines 119                    please write the formula using ‘Equation’ option in MS word (or similar option in other writing software), and the legends accordingly below it.

Comments on the Quality of English Language

English is written excellently with no apparent major grammatical flaw. However, font size and line spacing are different among paragraphs. Please set them consistently following the journal guidelines.

Author Response

Editor of Antibiotics                                                                    Wrocław, 20.12.24

Cover Letter

Dear Editor, we entirely agree with you that   "Self- citations may be valid if, for example, they are needed to understand the background and history of the work in question". This statement applies to our situation. We have been a pioneering phage therapy center in Europe with the largest number of patients treated and monitored for their antibody responses against phages. In fact, such responses are only rarely mentioned in publications related to phage therapy which often include only single cases. Moreover, our studies on factors influencing the intensity of antibody responses against phages during clinical phage therapy are quite unique. Therefore, when discussing the data presented in the current paper we had to refer to our earlier reports as no similar analyses had been published so far. Recent data derived from a murine model confirm our observations that two rounds of phage treatment are necessary to induce neutralizing antibodies (Berkson JD et al, Nature Com. 2024, 15:2993). This citation has now been added to the reference list. Our corrections are marked in red. Linguistic corrections from the reviewer 1 are marked in green

English has been checked again.

Regards

  1. Łusiak-Szelachowska

Reviewer 2

Thank you very much for your opinion and suggestions.

Our corrections are marked in red. Linguistic corrections from the reviewer 1 are marked in green.

We improved the methods, the results and the English.

Please add more details of the Results to the Abstract to improve its clarity and attract more Readers.

Done lines 21-25

Lines   65, 67 & 97     Please write city and country names where these institutions located as some Readers may not familiar with them. Please only use abbreviations (e.g. HIIET) after their first mention.

Done lines 70, 107

Table 1                        Why only rates of phage inactivation (K) before and during PT were presented here while patient bloods were also collected (and tested?) after PT?

We added K in tables after PT in some patients. The studies were conducted in accordance with the experimental therapy. We could not obtain the results after therapy from all patients.

Lines 91-93                 Please also provide these legends to Table 1 as it is a separate table to Table 2.

Done

Lines 105                    This statement needs reference(s). Done line 114

Lines 116-117             Please explain how to prepare the media agar, or at least its composition. Done lines 132-134

Lines 119                    please write the formula using ‘Equation’ option in MS word (or similar option in other writing software), and the legends accordingly below it. Done lines 135-136

English is written excellently with no apparent major grammatical flaw. However, font size and line spacing are different among paragraphs. Please set them consistently following the journal guidelines.

Done

Reviewer 3 Report

Comments and Suggestions for Authors

The authors report their recent findings on the levels of antiphage antibodies in patients undergoing phage therapy. Their main finding is the unexpected one that high levels of antiphage antibodies do not limit the effectiveness of phage therapy. Differences were found depending on the infections involved and the route of phage administration. Although the patient numbers involved are small, the study is interesting, consistent with other findings and should be of interest to the phage therapy community.

I have a few small points:

Line 37: the statement'  whereas local and both local and oral  administration are more immunogenic' was a little confusing; I think what is meant is 'the combination of both local and oral  administration...."

Table 1 was a little difficult to read by itself, without constant reference to the legend on the next page and the materials and methods section. Mostly, this is because of the numbers and letters used to denote the infection type and clinical outcome, respectively. I'd suggest that the authors dispense with at least one of these systems and simply use an abbreviation to denote infection and/or outcome.

line 102: virion, not virione

line 113 Peptone, not pepton.

Figure 2: the absence of a 'before' measurement with 2/Staph_1N needs a comment in the legend.

Can the authors comment on the host range of the phages used? Are they highly specific for particular strains, or not?

Author Response

Editor of Antibiotics                                                                    Wrocław, 20.12.24

Cover Letter

Dear Editor, we entirely agree with you that   "Self- citations may be valid if, for example, they are needed to understand the background and history of the work in question". This statement applies to our situation. We have been a pioneering phage therapy center in Europe with the largest number of patients treated and monitored for their antibody responses against phages. In fact, such responses are only rarely mentioned in publications related to phage therapy which often include only single cases. Moreover, our studies on factors influencing the intensity of antibody responses against phages during clinical phage therapy are quite unique. Therefore, when discussing the data presented in the current paper we had to refer to our earlier reports as no similar analyses had been published so far. Recent data derived from a murine model confirm our observations that two rounds of phage treatment are necessary to induce neutralizing antibodies (Berkson JD et al, Nature Com. 2024, 15:2993). This citation has now been added to the reference list. Our corrections are marked in red. Linguistic corrections from the reviewer 1 are marked in green

English has been checked again.

Regards

  1. Łusiak-Szelachowska

Reviewer 3

Thank you very much for your opinion and suggestions.

Our corrections are marked in red. Linguistic corrections from the reviewer 1 are marked in green.

Line 37: the statement'  whereas local and both local and oral  administration are more immunogenic' was a little confusing; I think what is meant is 'the combination of both local and oral  administration...."

Done line 42

Table 1 was a little difficult to read by itself, without constant reference to the legend on the next page and the materials and methods section. Mostly, this is because of the numbers and letters used to denote the infection type and clinical outcome, respectively. I'd suggest that the authors dispense with at least one of these systems and simply use an abbreviation to denote infection and/or outcome.

 Done. Instead of numbers, we have included types of infections in tables 1 and 2.

line 102: virion, not virione. Done line 112

line 113 Peptone, not pepton. Done lines 128-129

Figure 2: the absence of a 'before' measurement with 2/Staph_1N needs a comment in the legend.

Done. We included comment in the legend about the absence of antiphage antibodies in case 2/Staph_1N  before PT.

Can the authors comment on the host range of the phages used? Are they highly specific for particular strains, or not?

Done lines 115-120